

# Differential gene expression analysis by RNA-seq reveals the importance of actin cytoskeletal proteins in erythroleukemia cells

Vanessa Fernández-Calleja[1], Pablo Hernández[1], Jorge B. Schvartzman[1], Mario García de Lacoba[2] and Dora B. Krimer[1]

[1] Department of Cellular and Molecular Biology, Centro de Investigaciones Biológicas, Spanish National Research Council (CSIC), Madrid, Spain
[2] Bioinformatics and Biostatistics Service, Centro de Investigaciones Biológicas, Spanish National Research Council (CSIC), Madrid, Spain

Corresponding author
Dora B. Krimer, dbkrimer@cib.csic.es

## ABSTRACT

Development of drug resistance limits the effectiveness of anticancer treatments. Understanding the molecular mechanisms triggering this event in tumor cells may lead to improved therapeutic strategies. Here we used RNA-seq to compare the transcriptomes of a murine erythroleukemia cell line (MEL) and a derived cell line with induced resistance to differentiation (MEL-R). RNA-seq analysis identified a total of 596 genes (Benjamini–Hochberg adjusted $p$-value $< 0.05$) that were differentially expressed by more than two-fold, of which 81.5% (486/596) of genes were up-regulated in MEL cells and 110 up-regulated in MEL-R cells. These observations revealed that for some genes the relative expression of mRNA amount in the MEL cell line has decreased as the cells acquired the resistant phenotype. Clustering analysis of a group of genes showing the highest differential expression allowed identification of a sub-group among genes up-regulated in MEL cells. These genes are related to the organization of the actin cytoskeleton network. Moreover, the majority of these genes are preferentially expressed in the hematopoietic lineage and at least three of them, *Was* (Wiskott Aldrich syndrome), *Btk* (Bruton's tyrosine kinase) and *Rac2*, when mutated in humans, give rise to severe hematopoietic deficiencies. Among the group of genes that were up-regulated in MEL-R cells, 16% of genes code for histone proteins, both canonical and variants. A potential implication of these results on the blockade of differentiation in resistant cells is discussed.

## INTRODUCTION

Cancer cells are distinguished from their normal counterparts by several hallmarks, including uncontrolled growth, lack of response to apoptotic signals and blockade of differentiation (*Hanahan & Weinberg, 2000*; *Hanahan & Weinberg, 2011*). These characteristics serve as a framework for testing different protocols aimed at eliminating tumor cells by aggressive chemotherapy or radiotherapy. Alternatively, cancer cells may be

forced to resume the process of maturation by differentiation agents, which generally have less toxicity than conventional cancer treatments. An example of a successful clinical application of differentiation therapy is all-trans-retinoic acid (ATRA) for treatment of acute promyelocytic leukemia, which induces terminal differentiation of promyelocytic leukemic cells (*Nowak, Stewart & Koeffler, 2009*). Other differentiation-inducing agents, such as histone-deacetylase (HDAC) inhibitors (*Lane & Chabner, 2009*), cytidine analogs (e.g., 5′-aza-2′-deoxycytidine) (*Fenaux et al., 2010*), and tyrosine kinase inhibitors (e.g., imatinib) (*Haouala et al., 2011*) have been less successful in the treatment of leukemias and solid tumor cancers. An obstacle to all cancer therapies is the acquisition of drug resistance that develops in response to repeated therapies, which eventually leads to relapse in most patients (*Rebucci & Michiels, 2013*).

*In vitro* differentiation models have proved to be extremely useful to study the molecular events associated with the blockade of cell differentiation exhibited by some tumor cells and the requirements for re-entry into the cell differentiation program. The mouse erythroleukemia (MEL) model developed by *Friend et al. (1971)* is an outstanding example that remains as a solid platform to evaluate tumor cell reprogramming after more than 40 years since its description.

Friend erythroblasts are derived from mice infected with the Friend complex virus. Insertion of the Friend spleen focus-forming virus (SFFV) genome occurs several kilobases upstream of the *Sfpi1/PU.1* locus initiation start site (*Fernández-Nestosa et al., 2008*). This causes the constitutive activation of *Sfpi/PU.1* resulting in the blocking of erythroid differentiation and the development of erythroleukemia (reviewed in *Ruscetti, 1999*). MEL cells can be induced to reinitiate the differentiation program by the addition of chemical agents such as hexamethylene bisacetamide (HMBA) (*Fernández-Nestosa et al., 2008*). We have previously reported the establishment of an HMBA-resistant cell line (MEL-R) before. These cells were obtained after months of MEL cell culture in the presence of a differentiation inducer. The resulting cell line retained most of the native MEL cell characteristics. Unexpectedly, we found that *Sfpi1/PU.1* remains silent even though MEL-R cells do not differentiate, and this silencing persists in the presence of chemical inducers other than HMBA. Nevertheless, the SFFV integration site maps exactly to the same location both in MEL and MEL-R cell lines (2,976 bp downstream of the URE distal element). We also showed that inactivation of *Sfpi1/PU.1* in the resistant MEL-R cell line was mediated by DNA methylation at the promoter near to CpG islands (*Fernández-Nestosa et al., 2013*). For all these reasons, we believe MEL-R cells might constitute a useful model to study mechanisms that trigger inducer-resistant cell differentiation. Here we compared the differential expression profiles of MEL and MEL-R cells using RNA-seq to identify sequences potentially involved in the control of HMBA resistance. Our results revealed that a higher proportion of differentially-expressed genes are up-regulated in MEL cells than in MEL-R cells. Interestingly, a group of highly up-regulated sequences in MEL cells corresponded to genes encoding actin cytoskeleton proteins. A proportion of genes up-regulated in MEL-R cells belonged to histone coding genes. Canonical histone proteins H1, H2A, H2B, H3 and H4, are replication-dependent and their expression is coordinated with DNA replication, occurring primarily during the S phase of the cell cycle (*Rattray &*

*Muller, 2012*). There are nonallelic variants mainly of the H1, H2A, H2B and H3 histones that are not restricted in their expression to the S phase and have different physiological roles. Both groups, however, are essential elements of the nucleosome architecture and contribute to chromatin organization (*Talbert & Henikoff, 2010*). A potential contribution of histone gene expression to the differentiation block is also discussed.

## MATERIALS AND METHODS

### Cell cultures and treatment

MEL-DS19 cell line (hereafter called MEL) was obtained from Arthur Skoultchi (Albert Einstein College of Medicine, New York, USA). MEL-resistant cell line (hereafter called MEL-R) derived from MEL-DS19 was previously established in our lab by growing MEL cells continuously in the presence of 5 mM HMBA (*Fernández-Nestosa et al., 2008*). 3T3-Swiss albino fibroblasts cells were obtained from the Animal Cell Culture Facility from the Centro de Investigaciones Biológicas, CSIC. Cells were cultured in Dulbecco's modified Eagle's medium containing 10% fetal bovine serum, 100 units/ml penicillin and 100 µg/ml streptomycin (Gibco). Cell differentiation was induced by exposing logarithmically growing cell cultures to 5 mM HMBA. MEL-R cells were routinely cultured in the presence of 5 mM HMBA. Hemoglobinized cells were quantified by determining the proportion of benzidine-staining positive cells ($B^+$) in the culture.

### RNA isolation and RNA-seq analysis

Total RNA was isolated from $1 \times 10^7$ cells using the RNeasy kit (Qiagen). DNase I was used to degrade any possible DNA contamination. 1 µg of total RNA was used to prepare standard RNA-seq libraries (TruSeq RNA Sample Preparation Kit, Illumina) based on polyA$^+$ isolation. RNA concentration ranged from 326 to 394 ng/µl, and samples showed optimal integrity with RIN values of 9.80. The libraries had an average length of 337–367 nt and were quantified by quantitative PCR (Kapa Biosystems) using a previously quantified library as standard. Samples were loaded onto a lane of a flowcell using the Cluster Station apparatus (Illumina) and sequenced on the Illumina GAIIx platform (Parque Científico de Madrid, Spain) under a single read ($1 \times 75$) protocol. There were approximately 25 million and 17 million reads (75-nt length) for MEL and MEL-R libraries, respectively, which were used for further bioinformatics analysis. The median quality score was >30 across all sites of the reads (http://hannonlab.cshl.edu/fastx_toolkit/index.html). The sequencing reads were mapped to the mouse reference genome (Mus_musculus_NCBI_build37.67.cdna, 30-04-2012) with TopHat v2.0.1. TopHat's mapped reads were processed using the program Cuffdiff, a part of the Cufflinks software suite v2.0.0 (*Trapnell et al., 2012*) and measured as fragments per kb of exon per million fragments mapped (FPKM) (*Trapnell et al., 2012*). The workflow outlining the analysis of RNA sequencing data is shown in Fig S1. Samples (unreplicated MEL and MEL-R) were further analyzed using the DESeq package for R Statistical Analysis (*Anders & Huber, 2010*). We used DESeq as a complementary differential gene expression analysis to compare the results with those obtained by Cuffdiff (Fig. S2). DESeq was employed as an additional DEG calling method since it performs a different expression test (Fishers' exact test) than Cuffdiff ($t$-test). A list of differentially

expressed genes detected with both methodologies is included in Supplemental Information (Cuffdiff/DESeq analysis).

## Quantitative real-time PCR validation

Quantitative real-time-PCR (qRT-PCR) was used to validate the relative expression of genes selected from the RNA-seq analysis. Total RNA was extracted from $1 \times 10^7$ MEL and MEL-R cells as described above. In total, 2 µg of isolated RNA were transcribed to cDNA using random hexamers and 200 U of SuperScriptII Reverse Transcriptase (Invitrogen). Reactions were performed in triplicate using the SYBR Green Supermix (Bio-Rad) on an iQ5 System (Bio-Rad). The conditions for the amplification were as follows: pre-denaturing step of 95 °C for 3 min followed by 40 cycles of 9 °C for 30 s and 60 °C for 30 s, and a final ramp step of 1 °C/10 sec from 60 °C to 94 °C. The primer sequences were designed with Primer3 software (http://bioinfo.ut.ee/primer3-0.4.0/) (*Untergasser et al., 2012*) and are listed in Table S1 (for actin cytoskeleton genes), Table S2 (for histone genes) and Table S3 (for methylases and demethylases). Relative gene expression was analyzed by the $2^{-\Delta\Delta Ct}$ method as described in *Schmittgen & Livak (2008)*.

## Antibodies and western blotting

Control 3T3 fibroblast cells, MEL and MEL-R cells ($2.5 \times 10^6$) were harvested, washed with PBS and lysed with NP-40 buffer (20 mM Tris–HCl pH 7.5, 10% glycerol, 137 mM NaCl, 1% NP-40, 1 mM sodium orthovanadate, 10 mM sodium fluoride, 2 mM EDTA) containing protease inhibitors (all from Sigma). Protein lysates (10–30 µg) were separated by 12% SDS-polyacrylamide gel electrophoresis and transferred to PVDF membranes (Bio-Rad). The membranes were incubated with a mouse monoclonal anti-β-actin antibody (1:10,000, Sigma) and a rabbit polyclonal anti-α-tubulin antibody (1:1,000, ABclonal) followed by five washing steps with T-TBS (20 mM Tris–HCl, 150 mM NaCl, 0.1% Tween 20). Primary antibodies were detected by incubating with HRP-conjugated anti-mouse (1:3,000, Santa Cruz) or anti-rabbit IgG (1:1,000, DAKO) followed by five cycles of T-TBS washes. HP1 α (1:1,000, Millipore) and Sam68 (1:1,000, Santa Cruz) antibodies were used for Fig. S3.

## Bisulfite sequencing

The methylation analysis of *Btk*, *Plek* and *Was* promoter regions in MEL, MEL-R and differentiated MEL cells was performed using sodium bisulfite conversion. Genomic DNA from $8 \times 10^4$ cells was bisulfite-modified using the EZ DNA Methylation-Direct Kit (Zymo Research). Four microlitres of treated DNA was amplified by PCR using primers specific to the bisulfite-converted DNA for each promoter region with ZymoTaq DNA Polymerase (Zymo Research). The conditions for the PCR were as follows: pre-denaturing step of 95 °C for 10 min, followed by 40 cycles of 95 °C for 30 s, 55–60 °C for 40 sec and 72 °C for 40 s, with a final extension at 72 °C for 7 min. The primer sequences were designed using MethPrimer software (http://www.urogene.org/cgi-bin/methprimer/methprimer.cgi) (*Li & Dahiya, 2002*). The primers are listed in Table S4. PCR products were resolved in 1% agarose gels followed by sequencing for methylation analysis, which was performed by Secugen SL (CIB, Madrid).

## Cell cycle analysis

Cells ($2 \times 10^5$–$1 \times 10^6$) were harvested and fixed in 70% ethanol at 4 °C for 30 min. Fixed cells were washed twice in PBS and stained with propidium iodide/RNAse solution (Immunostep) for 15 min at room temperature (∼22 degrees Celsius). Cell cycle analysis was performed on a Coulter EPICS XL (Beckman) flow cytometer and cell cycle profiles were plotted using FlowJo software (Tree Star, Inc.) All samples were analyzed at least in triplicates for each experiment.

## Immunocytochemistry and confocal microscopy

Cells were plated on poly-L-lysine coated slides and incubated at 37 °C for 30 min. Cells were fixed with 4% paraformaldehyde for 30 min, permeabilized with 0.1% Triton-X 100 in PBS for 30 min and blocked with 1% bovine serum albumin in PBS/0.1% Triton-X 100 for 1 h, all at RT. Cells were stained with anti-β-actin (Sigma) or anti-HP1α (Millipore) antibodies, for 1 h at RT followed by washing twice with PBS. The primary antibody was detected with an Alexa Fluor 568 secondary antibody (Molecular Probes) and 1 µg/ml DAPI to stain nuclei, for 1 h at RT followed by two washes with PBS. Finally, cells were mounted on a cover slip with Prolong Diamond Antifade Mountant reagent (Invitrogen). Fluorescence images were acquired on a Leica TCS SP2 confocal microscope using a 100 × objective with zoom.

# RESULTS

## Differential gene expression between MEL and MEL-R

A total number of 25,791 genes were identified and deposited at the GEO database (GSE83567). 596 genes (Benjamini–Hochberg adjusted $p$-value $< 0.05$) were differentially expressed by more than two-fold between MEL and MEL-R cells, of which 486 genes were up-regulated in MEL cells and 110 were up-regulated in MEL-R cells. We focused our attention on sequences that were highly differentially expressed in MEL relative to MEL-R cells. Figure 1A illustrates the heat map that includes all the genes, clustered based on Ward's analysis of minimum variance criterion (Ward Jr, 1963), with fold change of 2 or more using Cuffdiff. An expanded heat map of genes showing highest fold-change values is shown in Fig. 1B. *Sfpi1/PU.1* was one of the selected genes that, as we demonstrated previously (Fernández-Nestosa et al., 2008), is not expressed in the resistant cell line and served in this case as a positive control for the RNA-seq efficiency. We observed that the majority of the differentially expressed genes are up-regulated in MEL compare to MEL-R cells (Fig. 1A and Fig S2).

Searching for common features among the cohort of highly expressed genes in MEL cells, we found that several of these genes were implicated in the regulation of the actin cytoskeleton organization. Table 1 lists the groups of these genes with the highest expression difference between MEL and MEL-R cell lines. In addition to their relationship with the actin pathway, eight of these genes (*Btk*, *Dock2*, *Itgb2*, *Nckap1l*, *Plek*, *Rac2*, *Was* and *Wdfy4*) were preferentially expressed in hematopoietic cells (www.proteinatlas.org) and at least three of them, *Was* (Wiskott Aldrich syndrome), *Btk* (Bruton's tyrosine kinase) and *Rac2*, when mutated in humans, give rise to severe deficiencies (Ambruso et al., 2000;

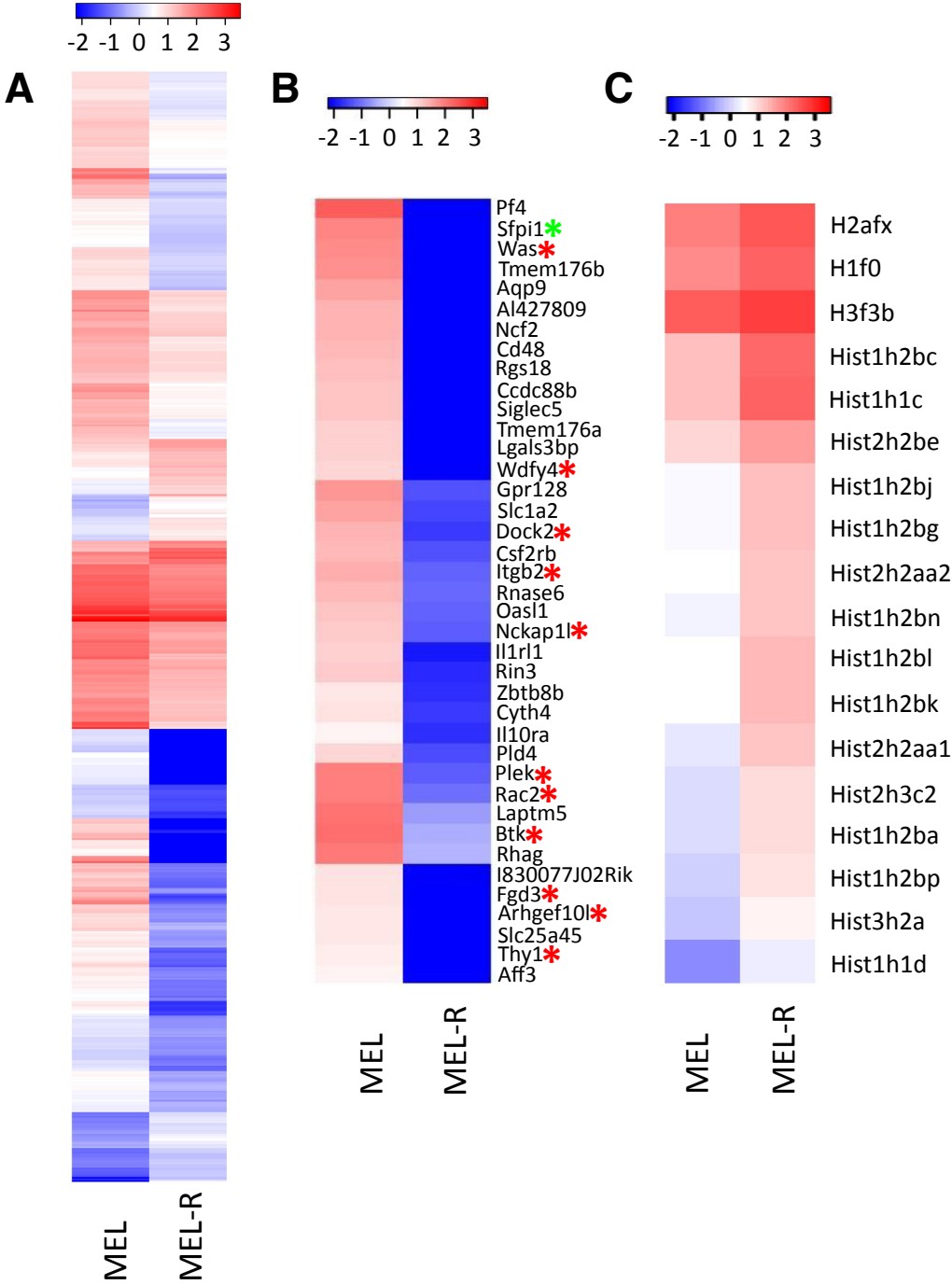

**Figure 1 Differentially expressed genes between MEL and MEL-R cell lines.** (A) Analysis of differentially expressed genes between MEL and MEL-R cell lines classified based on analysis of minimum variance (*Ward Jr, 1963*). (B) Heat map zoomed to amplify the genes with higher fold-change values. Genes related to the actin cytoskeletal network are indicated by red asterisks. As expected, *Sfpi/PU.1* presented strong differences in expression towards the progenitor cells and served as control for RNA-seq efficiency (green asterisk). (C) Heat map limited to histone gene expression. The color scales shown in all maps illustrate the $\log_{10}$ FPKM. Red and blue colors represent high and low expression, respectively.

**Table 1**  List of differentially expressed genes related to actin cytoskeleton.

| Gene | Locus | FPKM_MEL | FPKM_MELR | Log2 (Fold Change)[b] | FDR-adjusted $p$-value |
|---|---|---|---|---|---|
| [a]Plek | 11:16871208–16908721 | 90.85 | 0.06 | 10.56 | 0 |
| [a]Rac2 | 15:78389598–78403213 | 91.00 | 0.09 | 9.93 | 0 |
| [a]Dock2 | 11:34126863–34414545 | 23.21 | 0.03 | 9.74 | $1.08 \times 10^{-10}$ |
| [a]Btk | X:131076879–131117679 | 149.00 | 0.36 | 8.70 | 0 |
| Itgb2 | 10:76993092–77028419 | 27.24 | 0.07 | 8.61 | 0 |
| [a]Nckap1l | 15:103284255–103329231 | 13.91 | 0.05 | 7.98 | 0 |
| [a]Was | X:7658591–7667617 | 65.48 | 0 | 6.03 | $1.08 \times 10^{-6}$ |
| Wdfy4 | 14:33772732–33998252 | 9.78 | 0 | 3.29 | $5.65 \times 10^{-6}$ |
| Fgd3 | 13:49358478–49404577 | 6.88 | 0 | 2.78 | $7.64 \times 10^{-4}$ |
| [a]Arhgef10l | 4:140070399–140221820 | 6.28 | 0 | 2.65 | $2.65 \times 10^{-3}$ |
| Thy1 | 9:43851466–43856662 | 5.55 | 0 | 2.47 | $1.14 \times 10^{-2}$ |

**Notes.**
[a]Genes validated by RT-qPCR.
[b]In order to avoid in the log2FC calculation an infinite magnitude leading to indeterminate numerical results, the $\log_2 (1)$ is taken when the FPKM value is 0.

*Bosticardo et al., 2009*; *Conley et al., 2009*). The majority of these genes were mostly linked to the lymphoid or myeloid lineages, and several were reported in an erythroid context (*Schmidt et al., 2004*).

Among the 110 selected genes whose expression was higher in MEL-R cells than in the progenitor cell line, 16% encode histone proteins, mostly canonical but also variant histone types. An expanded heat map illustrating the differential gene expression of histones in MEL-R vs MEL cell lines is shown in Fig. 1C. The RNA-seq data revealed differences in the expression of histones that belong to canonical H1, H2A, H2B and H3 groups, and to the variant histones H1f0, H2afx and H3f3b. To understand the significance of the unexpected up-regulation of histone gene expression in MEL-R cells, we compared their DNA content with that of undifferentiated and HMBA-differentiated MEL cells by flow cytometry (Fig. 2). We found that the pattern of the major cell cycle phases, G1 vs S vs G2/M, was similar between MEL-R cells and undifferentiated MEL progenitors (MEL-0 h). By contrast, differentiated MEL cells (MEL-96 h) accumulated at G1, a phenomenon that has been previously observed during MEL cell differentiation (*Kiyokawa et al., 1993*; *Vanegas et al., 2003*; *Fernández-Nestosa et al., 2008*). Nevertheless, we observed that in terms of DNA content, MEL-R cells acquired a tetraploid phenotype as revealed by the shift in DNA content to the right (Fig. 2C).

## Validation of differentially expressed genes by qRT-PCR

To validate the results obtained by RNA-seq, we measured the expression fold changes of seven selected genes by qRT-PCR. Those genes (marked with "a" in Table 1), were selected as they showed highest differential expression values, were related to the actin cytoskeletal network and had hematopoietic specificity. RNA from MEL cells treated with 5 mM HMBA were included to allow comparison between the undifferentiated and differentiated MEL cells against the resistant MEL-R line. The expression patterns observed in all cases were consistent with the RNA-seq results (Fig. 3), confirming the near absence of expression of

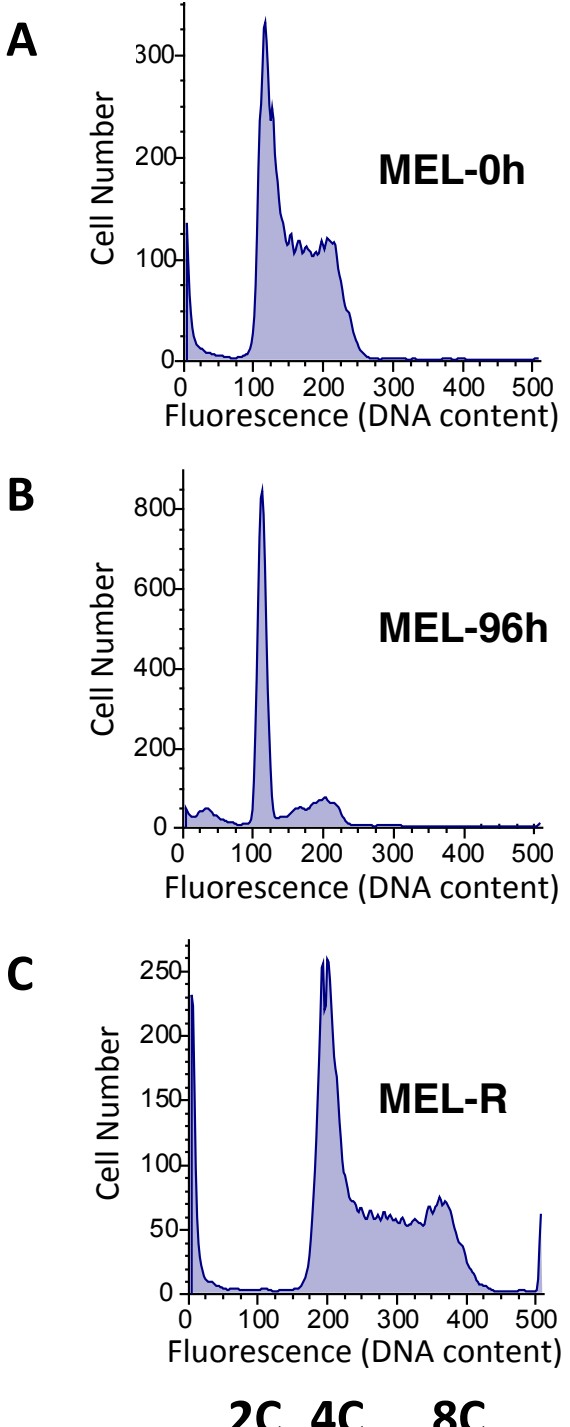

**Figure 2  Tetraploidy characterizes HMBA-resistant cells.** DNA content (2C, 4C and 8C) assayed by propidium iodide (PI) staining and flow cytometry show that (B) HMBA induced differentiated MEL cells (MEL-96 h) accumulate in G1 as compared with the (A) uninduced cell line (MEL-0 h). (C) DNA profile of HMBA-resistant cells (MEL-R) is similar to that observed in uninduced MEL cells regarding the fractions of cells in G1, S and G2-M. However, the DNA content profile is shifted to the right of the panel confirming that those cells become tetraploid.

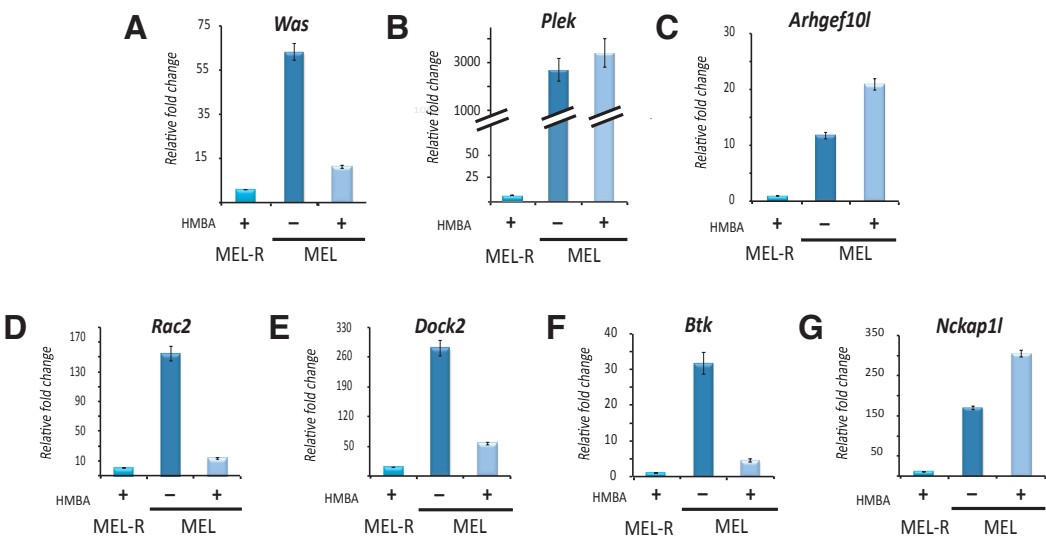

**Figure 3** **Validation of differentially regulated genes associated with the actin cytoskeletal network by qRT-PCR.** Selected genes that exhibited the highest FPKM values between MEL and MEL-R cell lines by RNA-seq were chosen for further validation by qRT-PCR. For the progenitor cell line, samples treated with HMBA for 96 h were also included. Data were normalized to $\beta$-actin expression for each sample. Bars represent ±SD of triplicate determinations ($P < 0.05$).

those genes in MEL-R cells. Significant differences were detected, however, when MEL-R cells were compared with MEL cells induced to differentiate with HMBA. Some of the genes such as *Was*, *Rac2*, *Dock2* or *Btk* shared a similar expression profile to that obtained in the resistant cell line, showing a tendency toward minimal expression, whereas the expression levels of *Plek*, *Arhgef10l* or *Nckap1l* exhibited either no change or a higher expression than that observed in differentiated cells.

Validation by qRT-PCR was also performed for histone genes and as before, we included a comparison with HMBA-differentiated MEL cells. The results of the qRT-PCR analysis were in agreement with those of the RNA-seq; in all cases, histone gene expression was higher in MEL-R cells than in MEL cells (Fig. 4A). The difference in the level of expression varied from more than ten-fold (*Hist1h2bk*) to two-fold (*Hist1h2bn*), except for *Hist1h2bj* with values close to 1. The same pattern was observed between the differentiated (MEL-96 h) and undifferentiated samples (Fig. 4B). These results ruled out the hypothesis that MEL-R tetraploidy was responsible for histone gene over-expression.

## Methylation status of CpG island promoters of Was, Btk and Plek

We have previously demonstrated that *Sfpi1/PU.1* silencing in MEL-R cells is caused by methylation of nearby CpG islands at its promoter (*Fernández-Nestosa et al., 2013*). Moreover, reactivation of silenced *Sfpi1/PU.1* occurs after treatment with 5-aza-2′-deoxycytidine, a potent inhibitor of DNA methylation. To investigate whether DNA methylation is responsible for the down-regulation in gene expression, we examined the methylation status of *Btk*, *Was* and *Plek* promoters in undifferentiated and differentiated MEL cells and in MEL-R cells by bisulfite sequencing. We mapped seven CpG islands upstream of the transcriptional start site of *Btk* and *Was* (Figs. 5A and 5B) and five in

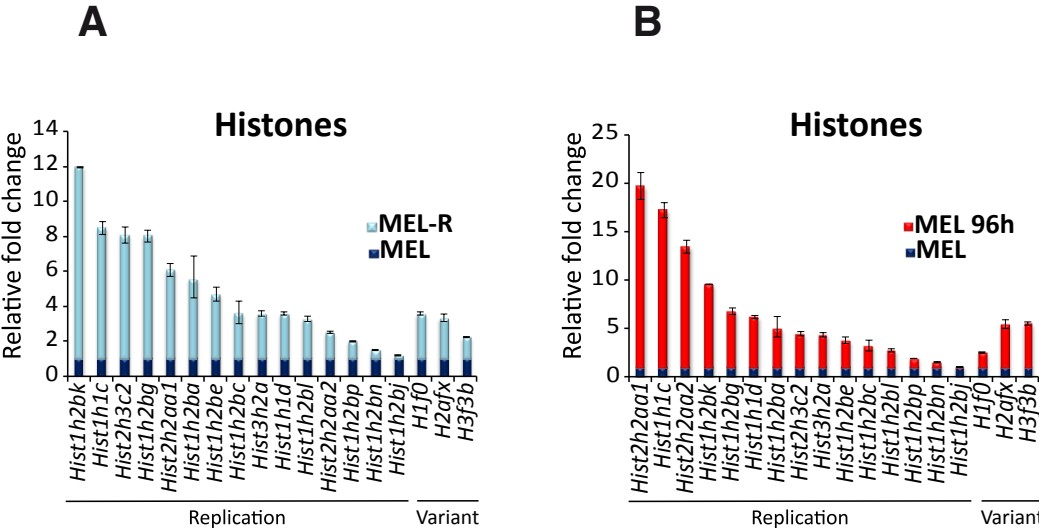

**Figure 4** **Differential histone gene expression between progenitor and resistant cell lines and after differentiation.** qRT-PCR analysis of histone genes, canonical and variant, up-regulated in (A) MEL-R cells relative to MEL cells, and (B) in HMBA-induced MEL cells (MEL 96 h) relative to uninduced cells (MEL). Data were normalized to $\beta$-actin expression for each sample. Bars represent ±SD of triplicate determinations ($P < 0.05$).

the case of *Plek* (Fig. 5C). Bisulphite sequencing revealed that all the CpG sites were hypomethylated in undifferentiated (0 h) and differentiated (96 h) MEL cells, whereas the promoters remained hypermethylated at all CpG sites in the resistant cell line. Sites 3, 4 and 5 at the *Btk* promoter were within a highly cytosine-rich region that were converted to thymine after bisulfite treatment, becoming difficult to resolve. We concluded from these experiments that *Btk*, *Was* and *Plek* expression was silenced by promoter methylation in MEL-R cell lines.

To confirm these results, we examined the expression pattern of the enzymes that catalyze DNA methylation (*Dnmt1*, *Dnmt3a* and *Dnmt3b*) and those that are involved in demethylation processes (*Tet1*, *Tet2* and *Tet3*). Quantitative RT-PCR analysis revealed that the level of expression of *Dnmt1*, the maintenance methylase enzyme, was higher in MEL-R cells than in undifferentiated or differentiated MEL cells, whereas the smallest changes were detected for the de novo methylases *Dnmt3a* and *Dnmt3b* between the different cell populations (Figs. 6A–6C). By contrast, expression of *Tet3*, but not *Tet1* and *Tet2* (enzymes involved in methyl group removal), was markedly reduced in MEL-R cells (Figs. 6D–6F). These results showed that the increase in DNA methylation by Dnmt1 in MEL-R cells overlaps with a decrease in demethylation by Tet3, which presumably results in the silencing of *Btk*, *Was* and *Plek* promoters.

## Actin cytoskeleton is poorly organized in resistant erythroleukemia cells

The actin cytoskeleton is composed of an extensive variety of actin regulators and nucleators that interact through a complicated protein network (*Moulding et al., 2013*; *Bezanilla et al., 2015*). Our analysis indicated that the expression of a group of genes related to actin

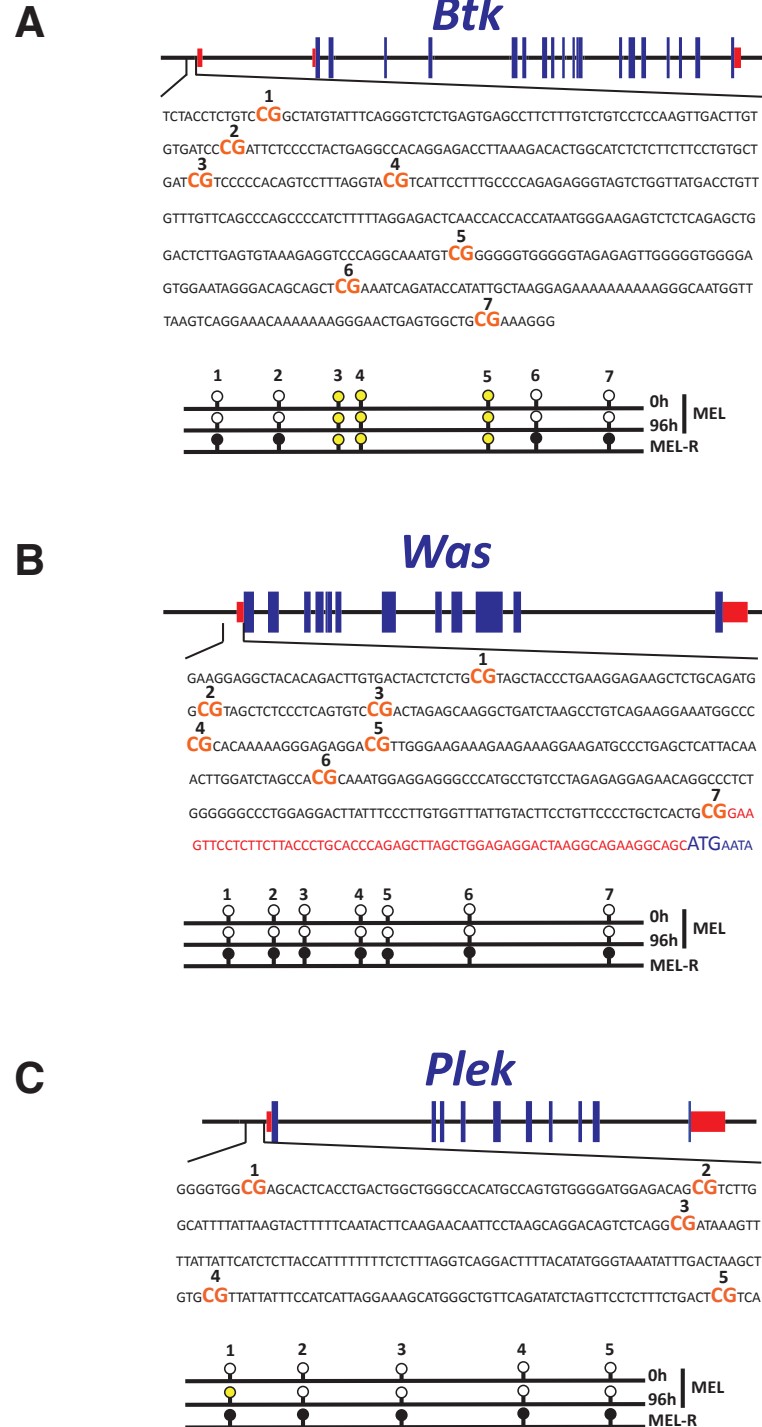

**Figure 5** **Methylation status of the Btk, Was and Plek promoters at the HMBA-resistant cells.** Genomic maps including exons (blue rectangles) and 5′ and 3′ UTRs (red rectangles) of (A) Btk, (B) Was and (C) Plek. Expanded regions illustrate the promoter regions containing seven CpG islands (CG) for Btk and Was and five CpG islands for Plek. "Lollipop" schematic diagram of methylation patterns is represented below each sequence. Results from untreated (0 h) or HMBA-treated MEL cells (96 h) as well as MEL-R cells are shown. Black and white lollipops indicate methylated or unmethylated CpGs, respectively, while undetermined methylation status (see text for details) is represented in yellow.

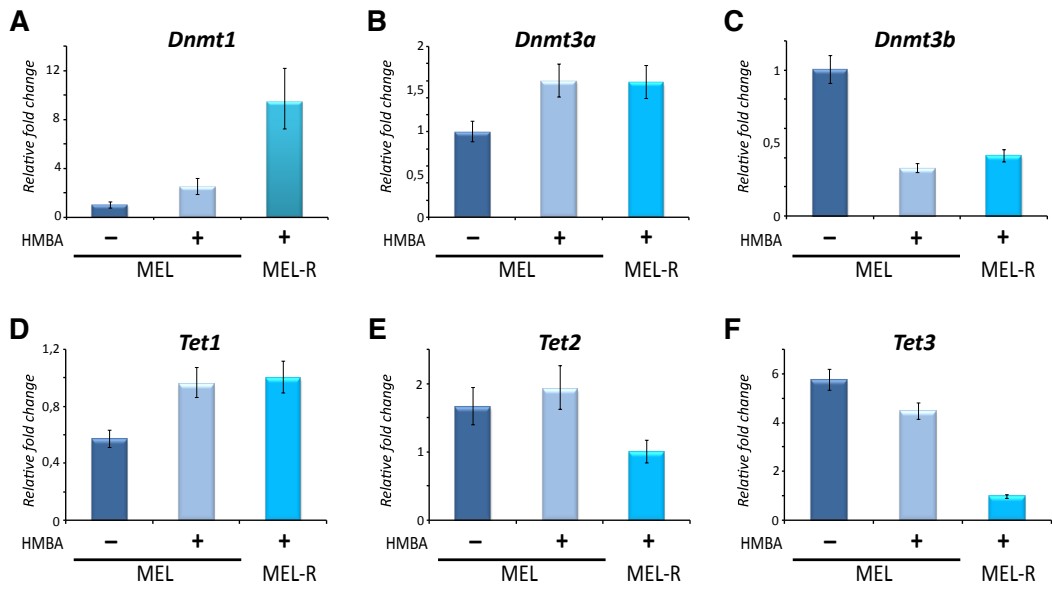

**Figure 6  High and low expression of Dnmt1 and Tet3, respectively, are related to gene silencing and DNA methylation in MEL-R cells.** qRT-PCR was performed for *Dnmt1*, *Dnmt3a* and *Dnmt3b* methylases and *Tet1*, *Tet2* and *Tet3* demethylases in undifferentiated, HMBA-treated MEL cells and in MEL-R cells. Data were normalized to $\beta$-actin expression for each sample. Bars represent ±SD of triplicate determinations ($P < 0.05$).

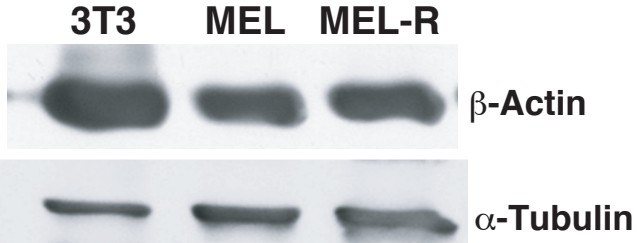

**Figure 7  Actin protein is equally abundant in progenitor and resistant MEL cells.** Western blot analysis for actin protein expression in MEL and MEL-R leukemia cells and in 3T3 control fibroblasts. Equal amounts of protein were loaded and immunoblotted with an anti-$\beta$-actin antibody. Anti-$\alpha$-tubulin was used as a loading control.

cytoskeleton organization was depressed in the resistant erythroleukemia cell line. To examine whether actin was affected by the silencing of genes related to actin polymerization and/or regulation, we evaluated its protein expression by Western blotting and found that its levels were similar between MEL and MEL-R cells (Fig. 7).

While these results demonstrate that the total amount of actin is equivalent for both cell lines, it does not reveal details of the actin organization. We therefore used fluorescence immunocytochemistry and confocal microscopy with anti-actin antibody to localize the protein in fixed MEL and MEL-R cells. In both populations, a rim of actin fluorescence was observed surrounding the nuclei (Fig. 8). However, MEL-R cells showed a significant reduction of signal intensity. These results were consistent with the RNA-seq analysis,

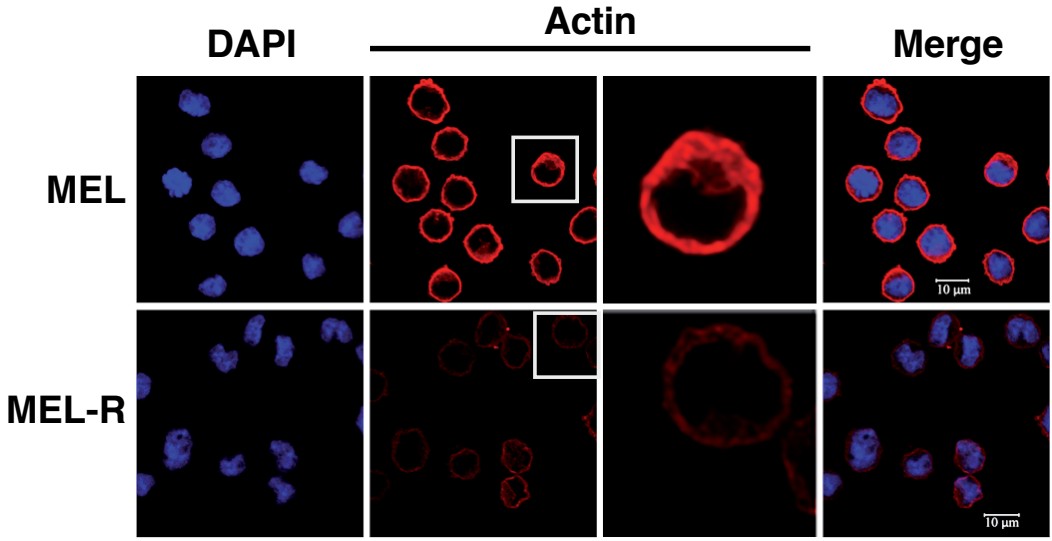

**Figure 8** **Actin cytoskeleton integrity is perturbed in MEL-resistant cell lines.** Confocal immunofluorescence microscopy of progenitor MEL cells and resistant MEL-R cells stained with a mouse monoclonal anti-$\beta$-actin antibody (red). Nuclear DNA was stained with DAPI (blue). Scale bar is 10 $\mu$m.

where a marked reduction in the expression of actin-regulators genes was detected in MEL-R cells, suggesting that actin cytoskeleton organization is perturbed in the resistant erythroleukemia cell line.

## DISCUSSION

### Actin-regulatory proteins are preferentially expressed in murine erythroleukemia cells

Cancer cells can acquire resistance to most traditional chemotherapy regimes and also targeted therapies, and such an occurrence remains a great concern in cancer treatment (*Raguz & Yague, 2008*; *Rebucci & Michiels, 2013*). Research on molecular and cellular mechanisms that confer resistance to tumor cells is therefore a major focus of basic and clinical investigation. Along this line, cell culture models have been crucial to advancing the understanding of cancer cell resistance. We took advantage of an HMBA-resistant cell line derived from murine erythroleukemia cells, previously established in our lab (*Fernández-Nestosa et al., 2008*; *Fernández-Nestosa et al., 2013*), to study the molecular events that contribute to the resistant phenotype. Both MEL and MEL-R cell lines are blocked at the proerythroblast stage of differentiation but unlike the progenitor cell line, MEL-R cells do not react to HMBA or other chemical inducers (e.g., DMSO, hemin and butyrate) and remain resistant against cell differentiation. In the present study, we used RNA-seq technology to identify genes potentially involved in the resistance mechanism. Our analysis identified 596 genes that were differentially expressed between progenitor and resistant cells, with the majority corresponding to genes up-regulated in MEL cells while only 110 were up-regulated in MEL-R cells.

Among these identified genes, some of them were prominent by their high expression in MEL compared to MEL-R and for sharing two important features: belonging to the actin regulatory network and being preferentially expressed in the hematopoietic lineage. Also, at least three genes, *Was, Btk* and *Rac2*, when mutated are linked to severe human hematological pathologies (*Ambruso et al., 2000*; *Bosticardo et al., 2009*; *Conley et al., 2009*). Additionally, a recent study showed that biallelic mutations in the *Dock2* gene result in severe immunodeficiency that leads to defects in actin polymerization (*Dobbs et al., 2015*). We hypothesized that proteins of the actin network such as Btk, Was and Plek among others described in Table 1, are essential for such organization although it is unclear whether the absence of expression is a cause or consequence of the defect.

### Transcriptional down-regulation of *Was*, *Btk* and *Plek* correlate with DNA promoter methylation

The network of actin filaments provides mechanical support to the cell cytoskeleton, but it is increasingly acknowledged that it also contributes to other critical cellular processes. Emerging evidence points to a role for the actin cytoskeleton in controlling and regulating receptor signaling (*Mattila, Batista & Treanor, 2016*). We show here a dramatic down-regulation of some of these network components in MEL-R cells, which show a relationship with the methylation status at nearby CpG islands in the promoters of *Was*, *Btk* and *Plek*. Over-expression of the methyltransferase Dnmt1, a maintenance methylase that acts on hemimethylated DNA, and the repression of the demethylase Tet3, supported these findings. These observations led us to speculate that silencing of most of the cytoskeleton-associated proteins is linked to a hypermethylation status. Interestingly, whereas no significant changes in total actin protein levels were observed between MEL and MEL-R cells, weaker signals were detected in MEL-R cells by immunocytochemistry. This observation might indicate poor actin organization. Regulation of actin polymerization in eukaryotes requires a large number of accessory proteins. These proteins facilitate polymerization or disassembly of monomeric globular actin (G-actin) into filamentous actin (F-actin) and vice versa. Many of these proteins interact with each other. For example, Btk interacts with Was and activates the protein by inducing its phosphorylation in B cells (*Sharma, Orlowski & Song, 2009*). Btk also promotes a Rac2 response, leading to F-actin rearrangements in mast cells (*Kuehn et al., 2010*). Dock2 is essential for lymphocyte migration and mediates cytoskeletal reorganization through Rac2 activation (*Fukui et al., 2001*). The transcription factor PU.1, responsible for the differentiation block in MEL cells but silenced in MEL-R cells, is a major regulator of Btk expression both in myeloid and lymphoid cells (*Himmelmann et al., 1996*; *Christie et al., 2015*). In summary, the actin cytoskeleton network is orchestrated by multiple associated proteins with possible overlapping roles, which contribute to different cell functions through complex associations. As we showed here, silencing of some of these proteins has deleterious effects on actin organization and we speculate that this might be a cause for the blockade of differentiation in resistant cells.

## Increased expression of histone-coding genes characterizes MEL-R cells transcriptome

As stated earlier, only 110 from 596 differentially-expressed genes were up-regulated in MEL-R cells. This indicates a tendency towards a general decline of gene expression in resistant cells, a situation comparable with what occurs during cell differentiation. Silenced compartments composed mainly of heterochromatin are considered hallmarks of the differentiated cells, a condition that progresses all through terminal differentiation (reviewed in *Politz, Scalzo & Groudine, 2013*). A gradual increase in heterochromatinization has been described in differentiating leukemia cells, as measured by the amount of the heterochromatin-associated HP1 α, which increases continuously during MEL differentiation (*Estefania et al., 2012*). Heterochromatinization is enhanced in MEL-R cells relative to undifferentiated MEL cells, but is nevertheless lower than in HMBA-differentiated cells (Fig. S3). The progressive gene silencing observed in MEL-R cells is one additional element that suggests that these cells are at a midway point between the undifferentiated and differentiated phenotypes due to a block somewhere in the process. Concomitant with this gene silencing, histone genes emerge as the major group up-regulated in the resistant phenotype. Initially, we associated the histone gene expression pattern with the tetraploid status of the MEL-R cell lines. Polyploidy has been reported in tumor cells as a result of stress-induced endoreplication (*Storchova & Pellman, 2004*; *Lee, Davidson & Duronio, 2009*). Chronic HMBA treatment might represent a hard-hitting stress that MEL-R cells overcome via a survival phenotype, i.e., tetraploidization, increased cell size and impaired cell differentiation. *Coward & Harding (2014)* in a comprehensive perspective support the hypothesis that tetraploidy provides numerous advantages during tumor initiation. Moreover, they present data supporting that polyploidy facilitates the acquisition of therapy-resistance in multiple cancers. MEL-R tetraploidy may possibly involve chromatin rearrangements and consequently histone gene expression changes. Nevertheless, the same fluctuations in histone gene expression were observed in differentiated cells, indicating that differentiated and resistant cells share a common mechanism not related to tetraploidy. *In vivo*, the quantity of reticulocytes, at a stage comparable to the last stages of HMBA-induced differentiation, increases several fold in a very short time (*Ji, Jayapal & Lodish, 2008*). It is speculated that a large number of histones needs to be generated. When reticulocytes mature, before enucleation, major histones are released into the cytoplasm through an unexpected nuclear opening that arises during terminal erythropoiesis. This migration is crucial for chromatin condensation and terminal differentiation (*Zhao et al., 2016*). We speculate that as an increase in histones occur both in HMBA-differentiated MEL and in MEL-R cells, a failure in chromatin condensation, either by an impairment in histone release or by a yet unknown mechanism, might interfere with terminal cell differentiation in resistant cells.

## CONCLUSIONS

A genome-wide RNA-seq analysis revealed that a subset of genes had significantly lower levels of expression in MEL-R cells compared to MEL cells. Among the differentially

expressed genes, a group up-regulated in the MEL cell line correspond to proteins related to the actin cytoskeleton organization. We showed here that the expression of these genes, i.e., *Was* (Wiskott-Aldrich syndrome), *Btk* (Bruton's tyrosine kinase) or *Plek* (Pleckstrin), among others, is very low in the resistant phenotype MEL-R. Immunocytochemistry and confocal microscopy analysis demonstrated an abnormal actin pattern in MEL-R cells, but the total amount of actin protein is equivalent for both, MEL-R and MEL. These results suggest that silencing of actin-related proteins influence the organization of the cytoskeleton. Among the group of genes that were up-regulated in MEL-R cells, histone proteins, both canonical and variants, were relevant.

## ACKNOWLEDGEMENTS

We acknowledge María-José Fernández Nestosa for her suggestions and critical reading of the manuscript, Alicia Bernabé for technical help and Eduardo Larriba for helping with the bioinformatics programs. We are grateful to the team from the Bioinformatics and Biostatistics Facility at the CIB for their efficient and dedicated technical support for the Next Generation Sequencing data analysis. We acknowledge Dr Tsung Fei Khang and two other anonymous reviewers as well as Dr Elena Papaleo acting as Academic Editor for their helpful comments that produced the final version of the manuscript.

### Funding

This work was supported by grant BFU2014 to JBS, PH and DBK from the Ministerio de Economía y Competitividad of Spain. The funders had no role in study design, data collection and analysis, decision to publish, or preparation of the manuscript.

### Grant Disclosures

The following grant information was disclosed by the authors:
Ministerio de Economía y Competitividad of Spain: BFU2014.

### Competing Interests

Dora B. Krimer is an Academic Editor for PeerJ.

### Author Contributions

- Vanessa Fernández-Calleja conceived and designed the experiments, performed the experiments, analyzed the data, contributed reagents/materials/analysis tools, wrote the paper, prepared figures and/or tables, reviewed drafts of the paper.
- Pablo Hernández and Jorge B. Schvartzman analyzed the data, contributed reagents/materials/analysis tools, reviewed drafts of the paper.
- Mario García de Lacoba analyzed the data, contributed reagents/materials/analysis tools.
- Dora B. Krimer conceived and designed the experiments, analyzed the data, contributed reagents/materials/analysis tools, wrote the paper, prepared figures and/or tables, reviewed drafts of the paper.

## Data Availability

The raw data files generated by RNA-seq have been deposited in the Gene Expression Omnibus (GEO) database http://www.ncbi.nlm.nih.gov/geo/query/acc.cgi?acc=GSE83567.

## Supplemental Information

Supplemental information for this article can be found online at http://dx.doi.org/10.7717/peerj.3432#supplemental-information.

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
