# Peer review of "Differential gene expression analysis by RNA-seq reveals the importance of actin cytoskeletal proteins in erythroleukemia cells"

_PeerJ, doi:10.7717/peerj.3432_

## Round 0.1 · original submission · Major Revisions

The manuscript has been evaluated by three expert reviewers. They all expressed major concerns on the manuscript as it currently stands. I would be glad to consider a substantially revised version of the work, including all the new analyses requested by the reviewers and where all their points will be addressed carefully.

Reviewer 1 ·

Basic reporting

*Main text*

The article is well written and clear, although I would suggest being more consistent in the terminology used. For instance, the MEL cell line, after being defined as such, is referred to as “MEL”, “MEL-DS19” (in the abstract), “progenitor”, and “parental”. Moreover, throughout the text, the words “genes” and “transcripts” are used interchangeably (e.g. reference to Figure 1 at lines 188 and 193). The latter issue prevent a correct interpretation of both the analytical methods and the results.

I report other minor comments in what follows:

Line 37: “A significant percentage (16%) corresponded to genes coding for…” should be replaced by “16% of genes code for…” (this percentage is nor large, neither significant).

Line 54: I would suggest using “solid tumors” as opposed to blood cancers.

Line 61: Please, write: “The mouse erythroleukemia (MEL) model”.

Line 66: Please, write the full name of PU.1 and Sfpi1 at their first occurrence in the text.

Line 69: The reader would benefit from a short explanation about HMBA effect on erythroleukemia cell lines to understand the properties of the resistant cell line.

Line 95: The sentence starting with “MEL-resistant…” seems incomplete.

Line 190: Please, write the number of genes in digits.

Line 196: “differential expression” is not a numeric entity and, thus, cannot be compared with a threshold of 2. Is this a fold-change or a log-fold-change? Is it considered as absolute value?

Line 212: Please, correct “a large proportion corresponded to genes encoding histone proteins” with “16% encode histone proteins”.

Line 234: Please, specify how these genes were selected among the others.

Line 238: I would suggest terminating this sentence as: “… confirming the near absence of expression in MEL-R cells for these genes”.

Line 275: I would not refer to the 2-fold change of Dnmt3b shown in Figure 7 as a “minimal change”.

Line 283: Please, change “extense” to “extensive”.

Line 293: Please, change “an antibody to actin” to “anti-actin antibody”

Line 297: Please, remove “strongly”.

Line 311: Please, refer to “murine” erythroleukemia cells.

Line 321: The meaning of “conspicuous” is not clear in this sentence.

Lines 322-32324: The theme of the “hematopoietic lineage” is repeated twice.

Line 333: Please, substitute “correlates well” as no correlation was assessed.

Line 352: “Hazard” is too strong. I suggest using “hypothesize” or “speculate”.

Line 358: A bracket is missing after the reference.

Line 373: Please change “sustaining” to “ supporting”.


*Literature*

I would suggest providing the references of all the computational tools cited in the methods (lines 95-96), as well as some references for the sentences of: line 54, line 351, and lines 214-221.


*Manuscript Format*

I would suggest revising the manuscript format according to the “standard sections” indicated in the “Instructions for Authors” (https://peerj.com/about/author-instructions/) because the “Conclusions” section is missing.

Moreover, please provide all necessary information about the computational analysis in the “Materials and Methods” section and not in the “Results” section (e.g. the information about the version of the reference genome used for read mapping should be reported at line 115 and not at line 186).


*Figures and Tables*

Table 1: I suggest adding the (adjusted) p-values for each gene. Moreover, it should be clarified whether these genes are those associated with actin cytoskeleton (as stated in the caption) or those with the highest expression difference (line 204). Also, please double check the data reported in this table and in the GSE83567 data, as they do not match. In the latter, for instance, the FPKM values in the two cell lines and the log2-fold-change for Was are, respectively: 0, 65.48, and 1.79769e+308.

Figure 1: this figure is not relevant to the analysis performed. Moreover, the caption is not clear (Are these genes or transcripts? Are this all genes or only the differentially expressed ones?) and the label on the x-axis is missing.

Figure 2: Please, specify the unit of measure of the color codes used and reported above the heat-maps and refer all of them to the same range so they can be compared. Moreover, I would not really describe a heat-map as a clustering analysis, especially in this case where each gene has only to values to be “clustered”.

Figure 5: This stacked bar graph is not very effective, because it is not clear what is the standard deviation bar for MEL. The representation of Figure 4 is clearer.

Figure 7: please, substitute “correlate” with another verb as no correlation is shown.

Experimental design

The analysis of RNA-seq data is hardly reproducible: its description is confused and incomplete and the different steps of data analysis cannot be reconstructed.

For instance, it is not described if and how the reads where filtered (line 112) and/or trimmed (line 185). The tools and the criteria used for read pre-processing should be described in the ”Materials and Methods”.

Moreover, for read mapping, authors should clarify whether TopHat was used with the default parameter settings or not.

The description of the analysis with the Cufflinks suite is also confused. The Cufflinks suite is composed of different tools to performs different tasks (http://cole-trapnell-lab.github.io/cufflinks/manual/), among which: quantification of transcript and gene expression with Cufflinks and differential expression analysis with Cuffdiff (I suppose these two tools were used given the GSE83567 data available). Moreover, transcript abundances are not an input of Cufflinks (line 186), but an output.

Another inconsistency regards differential expression: from the text is seems that differentially expressed genes were identified by simply putting a threshold on the expression fold-changes (estimated from Cuffdiff?). By applying this constraint on the GSE83567 data, more than 2800 differentially expressed genes can be identified, so I believe that p-values or adjusted p-values were also taken into account. From the abstract, it seems instead that differentially expressed genes are not identified by differential expression analysis with the Cufflinks suite, but by clustering. Finally, it is not clear whether and how DEseq tool was used, especially because DEseq cannot be applied to FPKM, but only to count data.

As, according to the title, differential expression analysis is the core of the article, the methods for this analysis should be explained in detail. This review might be useful to identify the different computational tasks to be performed:

Finotello, Francesca, and Barbara Di Camillo. "Measuring differential gene expression with RNA-seq: challenges and strategies for data analysis." Briefings in functional genomics 14.2 (2015): 130-142.

Validity of the findings

Differently from the qRT-PCR experiments, the design of the RNA-seq experiment presented in this study does not include any biological replicate and, thus, does not capture the biological variation in gene expression. As correctly pointed out in a comment by Hansen et al., despite its appealing low technical variability, RNA-seq cannot eliminate biological variability. Therefore, biological replicates are needed to going beyond the simple comparison of the two present samples.

Hansen, Kasper D., et al. "Sequencing technology does not eliminate biological variability." Nature biotechnology 29.7 (2011): 572-573.

I acknowledge, however, that the number of sequencing libraries impact on the experimental costs. If the aim of the study is to focus on the most sticking differences in gene expression, I would suggest increasing the number of replicates, but decreasing the sequencing depth through multiplexing.

Reviewer 2 ·

Basic reporting

The article is very well-written. Figures are relevant, clear and adequately described (see comments for some corrections) and the raw data is deposited to the GEO database.

Experimental design

The research is within PeerJ scope, the research question is well-stated. Materials and Methods are described with satisfactory detail.

Validity of the findings

There are some questions concerning statistics in some experiments (see comments). Conclusions are clear and relate to the original research question. There is some speculation in the discussion, this is however well identified as speculation.

Additional comments

1. Some points to the discussion: Immunocytochemistry of β-actin indicates that more protein is expressed in MEL cells, whereas in Western blot there is no difference. This discrepancy should be addressed in more detail. Is poor actin organization enough to explain the large difference in signal intensity? There is no indication that the antibody binds filamentous actin differently to globular actin? Also, a reassurance that the same parameters (exposure time) were used when imaging both MEL and MEL-R cells would help in assuring the validity of the findings. Also in lines 375-377, the fact that tetraploidy seems to not be behind increased histone expression could be emphasized similarly as in results, otherwise the long discussion about tetraploidy may overshadow this aspect.
2. A supplemental excel file where all the differentially expressed genes are listed could be added.
3. Some statistics concerning cell cycle distribution and tetraploidy would be needed (the number of replicates and whether the change was statistically significant).
4. Materials and Methods. Control 3T3 fibroblast cells should be mentioned in cell lines and growth media listed.
5. The naming convention of MEL cells should be made clearer in the beginning. I suggest that since in Materials and Methods, MEL-DS19 is indicated to be MEL in the following text, MEL is designated as MEL-DS19 at least in abstract, maybe in introduction as well. Especially in abstract (lines 31-33) it may be unclear whether simple MEL refers to MEL-DS19 or MEL-R (as there is yet no contrast or context to evaluate findings).
6. Table 1. The purpose of the red asterisk should be stated in Table legend (that those are the genes validated by qPCR). Otherwise it may seem that the asterisk indicates the same thing as in Fig 1, namely actin cytoskeleton-related genes (making the title seem confusing). In addition, the number of decimal digits is a bit too much, some rounding up should be done to make the table more readable.
7. Some typos and other writing errors. Line 95: The sentence has no verb (or is derived intended as the verb?) and feels disjointed. Modify e.g. in the following way: “HMBA-resistant MEL cells (hereafter called MEL-R) derived from 96 MEL-DS19, were previously established in our lab”. Line 196, illustrates, line 283, extensive, line 310 would be better as crucial to advancing the understanding…, line 358, reviewed in [33]), line 373, polyploidy).
8. There is inconsistent use of city/country information when mentioning companies, please check that all have the information (e.g. Gibco in Cell cultures and treatment, line 98) or the information is omitted from all companies. For consistency, Coulter XL flow cytometer and FlowJo manufacturers (lines 159-160) should be mentioned (even though they may be apparent from the name).
9. Results (Differential expression between MEL and MEL-R): Perhaps some of the content describing the method could be moved to Materials and Methods.
10. Figure 1 Legend. Some typo or something else going on (“FPKM MEL &λτ”)? Should the axis have some kind of title?
11. In the results section, references could added to the part about histone variants (line 217).
12. Lines 54-57: The sentence could be interpreted so that combination treatments are offered after relapse (when in the study cited, the combination therapy was offered to patients that had no prior therapy). The sentence could be clarified with this in mind.
13. Page numbers seem to be missing (they are perhaps added later?).
14. Citation style. In-text citations seem to be preferably written with citing names (not assigning numbers) in other articles published in PeerJ.

·

Basic reporting

In general, the usage of language is good but the manuscript requires reorganization for clarity.

1. I have the impression that part of the reporting in the result section should really belong to the materials and methods section, and some of the reporting in the results section rightfully belongs to the discussion section. Reorganizing these sections would certainly improve the reading experience. I provide some suggestions here:

(i) L.181-188 describes the methodology of RNA-Seq used, it shouldn't belong in the Results section.
(ii) L.205-209 describes the context of the selected genes in actin cytoskeleton organization and should be in the Discussion section.
(iii) L.213-224 described the role of histones - see how this information can be integrated into the Introduction section or the Discussion section.
(iv) L.240-246, L.299-301 contains speculative material and should be moved to Discussion section.
(v) L.283-287 contains description of actin cytoskeleton and is best moved to the Introduction section. Also, please avoid exaggerated descriptions such as "was profoundly depressed" in scientific writing.

2. The Introduction section can be written in a more focused way with respect to the authors' results that are based on a mouse model of leukemia. I feel that the present description about what cancer is (L.44 - 57) and what treatments are available for it, is a cliché. Giving the background about leukemia, and how a mouse model of leukemia have yielded insightful results leading to advances in the understanding of pathogenesis and treatment options, would motivate the reader much better than the current set up. Furthermore, introducing the role of genes such as Btk, Rac2, Was, Plek, and the histone genes in this section would help readers grasp the significance of the authors results later. In the revision, it would also be good if the authors can explicitly state the objectives and hypotheses being tested in the study.

3. The Discussion section would benefit from having subsections focusing on different aspects that the authors wish to discuss in relation to their experimental results. As it is, I find it rather challenging to digest the contents of the Discussion section. A final paragraph summarising the "take home message" will be much appreciated by the reader.

Minor specific comments:

L.46 "aimed to eliminate" should be "aimed at eliminating"

L.66 "Insertion of the Friend spleen focus-forming virus (SFFV) several kilobases" - does this refer to the entire virus genome or a particular gene in the virus's genome? Please clarify.

L.67 "resulting in a block of" should be "resulting in the blocking of"

L.80-88 "Our results ... is also discussed." The sentences describe the experimental results and interpretation, and should not be put in the Introduction section.

L.145 "performed by sodium bisulfite conversion" - should be "performed using sodium bisulfite conversion".

L.366 What is "genetic blackout"?

Either omit Fig.1, or replace with box plots.

Fig.2 - I think red and green colors are not color-blind friendly as per PeerJ's color publication of policy and should be changed. The blue - white - red color tone is a useful substitute. Please make clear whether the color bar represents the Z-score.

Fig.3 - Please state what "2C","4C" and "8C" mean in the caption, and also please label the x-axis clearly.

Table 1. Please, do not use so many significant digits for the results of FPKM and fold change. Reporting two is sufficient.

Experimental design

The authors address the question of whether genes involved in actin cytoskeletal organization are associated with the development of differentiation resistance in a mouse model of leukemia. However, in its present state I find gaps in the description of the experimental design that prevent confident acceptance of the authors' results.

1. There is no mention of the number of replicates for RNA-Seq experiment. One might deduce, from the heatmap in Fig.2A (which the authors describe as being produced using the ward.D algorithm - please state the gplots R package and mention that hierarchical clustering of the genes was done), it appears that the experiment is unreplicated (n=1 for MEL-R, MEL). Could the authors have done pooling? Please clarify, as different differential gene expression analysis programs vary in their effectiveness in the analysis of RNA-Seq data from unreplicated experiments.

2. For qPCR, it is mentioned in the caption of Figure that triplicates were used, but please make this explicit in the main body of the text in the materials and methods section.

3. It appears to me that the authors simply used a fold-change cut off criteria to select the differentially expressed genes, without considering the statistical significance. And then, among this set of hundreds of genes, they selected gene sets with biological functions which they believe can explain the observed variation in differential expression between MEL and MEL-R cell lines for further qPCR-validation. While the qPCR-validation is ultimately the gold standard in deciding whether an observed fold-change difference is believable, many would consider the authors' RNA-Seq analysis to be suboptimal.

Firstly, if they do not plan to use statistical significance, one can simply use Excel to find out the genes with fold-change exceeding a certain cut-off, and hence the reason for running DESeq is unclear. It is also a rather inefficient (screening through hundreds of genes) and potentially biased way to pinpoint gene subsets for further investigation.

A more reasonable way to discover differentially expressed gene sets is to select genes based on their fold change and p-values, and to perform differential gene expression analysis using several commonly available programs such as DESeq2 (the updated version of DESeq), edgeR, voom, NOISeq, GFOLD, etc. Some of these programs return smaller gene sets that have high potential to be q-PCR validated (e.g. GFOLD), while others control false discovery rates better (e.g. looking at gene sets common to edgeR and DESeq2). Once the authors have some confidence in the differentially expressed gene set returned from these programs, running DAVID to perform functional enrichment analysis would not only confirm the functions hypothesised by the authors to be related to differential expression, but also potentially uncover unsuspected functions. It would also be good if the authors could check the KEGG pathway to see what sort of pathways their differentially expressed genes are involved in. Please note that the effectiveness of these programs depend on whether replicates are available or not.

Minor specific comments:

L.110 "using a previously quantified library as standard" - Please provide citation.
L.110 "Samples were loaded onto a lane of a flowcell..." - Do the authors mean "lanes"?
L.112-114 Some important information are lacking. For example, what is the average read length (before and after trimming)? What was the quality score cut-off used? What was the trimming program used (with trimming parameters)? Can the authors provide the quality control diagnostics (the standard FastQC boxplots) for checking? Description of the outcome of quality control appears to be missing in the Results section concerning RNA-Seq analysis.
L.157 "x" and the multiplication symbol are confounded
L.159 "at room temperature" - What is this value (approximate range is OK) in degrees Celsius? Please be precise. Room temperature in different countries / seasons means different things.
L.160 What is the citation for "FlowJo" software?
L.171 "using a 100x objective and zoom" - Is it "a 100x objective lens"?

Validity of the findings

The lack of adequate explicit descriptions in the experimental design section encourages doubt about the interpretations of results made by the authors.

1. L.37-38 "Among the group of genes that were up-regulated in MEL-R cells, a significant percentage (16%) corresponded to genes coding for histone proteins" - I am not sure if "significant" is warranted here, as it can be misread as implying statistical significance. Perhaps "non-trivial" is safer. It is probably better to state explicitly the ratio of histone genes to the total number of up-regulated genes. Note that the percentage is sensitive to the program used to call differentially expressed genes, since different programs will report different number of genes up-regulated in MEL-R genes.

2. L.193 "Overall, the total number of genes expressed in the parental cell line decreased as the cells acquired the resistant phenotype". This conclusion may be methodology-dependent. I think it is only safe to say so if the authors consistently observe this pattern when different differential gene expression analysis programs are used to perform RNA-Seq analysis.

3. L.208-209 "The majority of these genes were mostly linked to the lymphoid or myeloid lineages" - This statement is unclear - what does "linked" mean? Absence/Substantially reduced of expression in lineages other than the two stated?

4. L.252 "although the difference in the level of expression varied from ... to two-fold (Hist1h2bj)" - This statement is somewhat inaccurate, as Hist1h2bj expression is very close to 1.

5. L.354-356 Again, if one is to depend on the computational result from RNA-Seq analysis for this conclusion, then checks using different differential expression analysis programs should be made to see if this general pattern holds.

Additional comments

I have reviewed the paper primarily from a scientific and methodological perspective. As I am not a biologist specializing in blood disorder, my knowledge of the biological interpretation of the result, (and also the literature in blood disorder) will be limited to whatever biological knowledge that I happen to have from experience. A non-trivial amount of additional work is needed to address the issues raised above, but I believe it can be done. Please ask the editor for more time to complete the revision.

---

## Round 0.2 · Major Revisions

The manuscript has been substantially improved but there are still some major concerns in the presentation of the data, some of the analyses and the suggestion to reorganize the manuscript to achieve a better flow. I would thus recommend the authors to address in details all the points that have been raised by the two reviewers, who carefully read and nicely evaluated the revised manuscript.

Reviewer 2 ·

Basic reporting

Reordering of some of the contents has made the flow better.
There are still some typos and errors (line numbers as in the track change file):
Line 35: related to the organization
Line 94: Significant should be removed
Line 131: “There were approximately 25 million and 17 million reads…”
Line 243: strengthening
Line 326: the smallest
Line 345: with anti-actin antibody to
Line 378: three genes, Was, Btk and Rac2
Line 381: result in severe
Line 452: provide
Line 461: silencing of actin-related

In addition, in Fig 1, panel A, the scale is unbalanced, going from -2 to 3.
In S2 Fig legend the use of contig is ambiguous. For example, “higher contig values” probably means just higher expression?

Experimental design

The additions to Materials and Methods clarified especially the sequencing-related methods. The section is now more unified.

Validity of the findings

The questions regarding statistics have been answered.

Additional comments

The data has been deposited in GEO. In order to make the data more accessible to a regular biologist, an Excel with all the differentially expressed genes could be added. However, the authors felt this was redundant. Still, a lot of biologists without proper tools may be discouraged from looking for further data due to lacking the relevant programs and knowledge (thus the authors may even miss a citation!). However, this is only for consideration.

·

Basic reporting

There is quite a large number issues which the authors need to address to improve the overall readability of the manuscript.

1. Having understood the work a bit better, I now think that the title “Differential Gene Expression Analysis by RNA-seq Reveals the Importance of Actin Cytoskeletal Proteins in Leukemia Cells” could be made more specific. Personally, I think "Reveals the Importance ..." is vague (what aspect?), and "leukemia cells" should be qualified as "a murine model of leukemia" to be precise.

2. The authors have a penchant to write paragraphs that are very long, such as L.65-L.96, L.351-375, and L.377-410. This will do them no good, because such paragraphs prevent meaningful retention of any points that the authors wish to make. For starters, try starting new paragraphs at L.360, and L.371. I insist that they do this to make their work more readable.

3. I still find quite a few instances where material that should be in the methods section are presented in the results section (e.g. L.276 – 282). Please check and reorganise. The results section should be clear and direct so that readers will grasp the findings easily.

4. I think the authors cited the wrong reference for DESeq, which is a highly cited paper with over 5000 citations. It is not Anders et al. 2012, as given by the authors. See https://www.ncbi.nlm.nih.gov/pubmed/20979621 for the correct one. Also, I believe Paul et al. (1991) is a more appropriate citation here than Fernandez-Nestosa et al. (2013) (L.68). (see Paul et al. 1991. The Sfpi-1 proviral integration site of Friend erythroleukemia encodes the ets-related transcription factor Pu.1. Journal of Virology, 65: 464-467.)

5. When multiple references are cited, it is better to sort them by chronological order, rather than alphabetical order (e.g. L.243 - (Fernandez-Nestosa et al. 2008; Kiyokawa et al. 1993; Vanegas et al. 2003).

6. Please check the abstract again on page 4 of 41. I still find “a significant percentage (16%) corresponded to genes coding for histone proteins” although this has been corrected as explained by the authors (page 6 of 41).

7. Table 1 - Ranking the genes by alphabetical order does not make much sense. Please rank them by log2(fold change). For p-value, only the order of magnitude is important. Also, I think the locus position information for the genes can be omitted, retaining only chromosome number.

8. If the indicator bar indicates log (is it base 10?) FPKM (Fig.1), then the caption of the figure should clearly indicate that, because Z-scores are commonly used in heatmaps.

9. Accents in non-English language should be consistently applied in text and references (e.g. Fernández-Nestosa not Fernandez-Nestosa).

10. Shouldn’t italics be used for gene names in murine models?

11. Be sure to change all instances of “Sx Table” to “Table Sx”.

12. Please provide information about the number of rows (genes) in the count table (it seems to be ~25000 based on the GSE data set that they deposited in NCBI).


To address the authors’ points in their rebuttal letter:

A1. “The color bar represents log FPKM, as already mentioned above for reviewer 1, and for clarity this information is added in the legend of Figure 2 (now Figure 1). We agree that red and green colors are not color-blind friendly and will be seriously considered for future illustrations.”

I think, it is better that the authors correct the colours in the plots now, rather than during the copyediting stage, which I am pretty sure they will be asked to based on my experience with dealing with PeerJ production team. I hope the authors will make this effort now. The issue of the need to respect the disability of colour-blind scientists (there are quite a few of them around) has been discussed many years ago in Nature blog (see http://blogs.nature.com/nautilus/2007/02/post_4.html).


A2. "Although room temperature in different countries means different things, room temperature in molecular biology laboratories differ very little. It is commonly accepted in most articles and usually means (approximately) 22 degrees Celsius."

Very well, then the authors should just be explicit by adding "(~22 degrees Celsius)" after "room temperature" (L.176)

A3. “This is a standard way to mention a 100x objective.”
I think the authors are trying to say "100x objective with zoom".


Minor corrections:

L.25 - “of an erythroleukemia cell line” should be qualified as “of a murine erythroleukemia cell line”.

L.27 - “RNA-seq analysis identified a total of 596 genes with a p-value adjusted less than 0.05” – it is better to write “RNA-seq analysis identified a total of 596 genes (adjusted p-value < 0.05)” - Here, the authors need to check whether Benjamini-Hochberg correction was used. If so, say “(Benjamini-Hochberg adjusted p-value < 0.05)”.

L.54 - “to all cancer therapy” – “therapy” should be “therapies”.

L.54 - “in the treatment of leukemias and tumors” – should be “leukemias and solid tumor cancers”; I think we treat cancers, not tumors.

L.54-55 - “… is the acquisition of drug resistance that develops in response to repeated therapy and inevitably leads to relapse in most patients” – should be “… is the development of drug resistance in response to repeated therapies, which eventually leads to relapse in most patients”.

L.67 - “Insertion of the Friend spleen focus-forming virus (SFFV) several kilobases upstream of the Sfpi1/PU.1 locus initiation start site leads to its constitutive activation,”

The authors should mention that it is the genome of the virus that is being inserted. Also, it is better to replace “leads to its constitutive activation” with “causes the constitutive activation of Sfpi1/PU.1”, since the position of the pronoun “its” in the sentence can ambiguously refer to the viral genome.

L.69 - “by the addition of chemical agents such as HMBA” – declare the full name of HMBA first before using it.

L.81-96 – The authors included results in the introduction section, which is inappropriate.

L.105 – “Fibrobasts 3T3-Swiss albino were” should be “3T3-Swiss albino fibroblasts cell lines were” .

L.102-103 - “MEL-DS19 (hereafter called MEL) were obtained …” ; “MEL-resistant (hereafter called MEL-R) derived …” – add “cell lines” after “MEL-DS19” and “MEL-resistant”.

L.104 – “by culturing during minimum 12 weeks in …” – sentence has grammar error.

L.110 – “the differentiation inducer” – be explicit about what this inducer is.

L.110 – “monitored” should be “quantified”.

L.113 - “RNA isolation and RNA-seq” – should be “RNA isolation and RNA-seq analysis”

L.114 - “Total RNA was isolated from 1×10^7 cells using ..” - How do the authors know that there were 10 million cells?

L.115 – “1 ug was” – 1 ug of what? (see L.136).

L.122 – “Reads were approximately 25 million and 17 million for MEL and MEL-R libraries, respectively” should be “Reads (75 nt) …”

L.126 – “Mus_musculus_NCBI build 37.2” – Is this correct? When I checked the metadata of the GSE data deposited by the authors (GSE83567), it shows that the transcripts were mapped to “mouse cDNA Ensembl (NCBIM37.67.cdna, 30-04-2012) using BWA, Release 0.6.1 with parameters aln -t -o 2 -e 2 -1 30 -k 3”.

L.144 – “as described (Schmittgen & Livak 2008).” – should be “as described in Schmittgen & Livak (2008).”

L.161 – “promoter” should be “promoter”. There are a few more such instances which the authors should seek out and correct.

L.161 – “The analysis of Btk, Plek and Was promotor regions in MEL, MEL-R and differentiated MEL” - Please be specific about the type of analysis, which is differential methylation analysis.

L.163 - “bisulfite modified” – link the two words with a hyphen.

L.193 – Add “available at” before the URL.

L.199-202 – These sentences have no place in the results section and are best omitted.

L.207-209 – “classified based on analysis of minimum variance (Ward 1963), with a differential expression greater than 2-fold using Cuffdiff.” – should be “clustered based on Ward’s analysis of minimum variance criterion (Ward 1963), with fold change of 2 or more using Cuffdiff.”

L.223, L.199 – Why use “sequences” instead of “genes”?

L.225 – “a good number of these genes were specific to the hematopoietic lineage” – please be precise – “a good number” is subjective.

L.229-230 - “… and fewer were reported in an erythroid context.” – should be "… and several were reported in an erythroid context."

L.245 – “(Fig 2, bottom panel)” – Please be specific – is it subpanel A, B, or C?

L.249 - “Validation of RNA-seq data by qRT-PCR” – to be precise, the authors validated differentially expressed genes; data cannot be “validated”.

L.251 – “Those genes, marked with a red asterisk in Table 1, showed highest differential expression values, are related to actin cytoskeletal network” – please rephrase for grammatical correctness.

L.256 – “the near absence of expression in MEL-R cells for those genes” – should be
“the near absence of expression of those genes in MEL-R cells”.

L.315-316 - “a rim of actin fluorescence was apparent surrounding nuclei (Fig 8)” – please correct the grammar.

L.381 - (reviewed in (Politz et al. 2013)) – double parentheses.

L.395 - “Coward and Harding in a comprehensive perspective support the hypothesis that
396 tetraploidy provides numerous advantages during tumor initiation. (Coward & Harding 2014).” - should be “Coward and Harding (2014) in a ….tumor initiation.”

L.401 - “share a common mechanism, not related to tetraploidy” – remove comma.

L.415 - “MEL-R cells provides a useful” – subject-verb agreement problem.

L.425-426 – “both canonical and variants were relevant” - add a comma after variants.

L.434 - “grateful the team from” – add "to" after "grateful".

Experimental design

This part is now generally better but a few issues remain.

1. I think details of the read preprocessing is unclear. From the quality score box plots provided in the rebuttal letter, it seems that median quality score was indeed > 30 across all 75 sites, so no trimming was necessary. Make it clear. The authors should explicitly mention that Cuffdiff and DESeq were used to perform differential gene expression analysis, not just "analysis" or "being analyzed". For unreplicated RNA-Seq experiments, normalisation has little effect on the outcome of the analysis.

To answer the authors’ replies in the rebuttal letter:

B1. "Samples for MEL and MEL-R can be express as "pooling" if considered as such each cell culture composed by millions of cells. Although the experiment is unreplicated, the data analyzed with Cuffdiff and DESeq follow the same pattern, strengthening RNAseq data. Ultimately, all these results were strengthened by the qRT-PCR analysis."

As the authors themselves say so, their experiment is unreplicated. This itself is not a problem, but the authors need to explicitly mention it in the methods section for the purpose of clarity in explaining their experimental design.

B2. "We used DEseq as a complementary analysis to compare the results with those obtained by Cuffdiff. We observed that the results in both cases follow the same pattern, strengthen RNAseq data. We added a paragraph in the "Results" section (L208) and a Supplementary Figure (S2 Fig.) with the DEseq results."

It is good that the authors checked the differential expression results obtained using CuffDiff with DESeq. However, the description of the new result is unclear. The authors said that “that the results in both cases follow the same pattern, strengthen RNA-seq data (S2 Figure).” However, while the actin-associated genes in the list of differentially expressed genes with high fold change are almost the same, the overall pattern is different (compare leftmost panel of Figure S2 with Figure 1). Generally when comparing the list of differentially expressed genes from several different programs, one uses the Venn diagram to visualise genes that are common for different methods, and unique to each method. I think this information is missing in the current revision.

Minor points:

L.102 - “MEL-DS19 (hereafter called MEL) were obtained from Arthur Skoultchi (Albert Einstein College of Medicine, New York, USA).” - Please provide the time when the cell lines were obtained.

L.177 – “DNA content was analyzed with FlowJo software” - What are these analyses? Cell cycle analysis? Proliferation analysis? Ploidy analysis? Please specify.

Validity of the findings

There remain several statements which lack support from evidence.

1. In L.215-220, the authors wrote:

“Although other methods originally programmed to deal with unreplicated RNA-seq data were described ((Khang & Lau 2015), and references therein), our data showed a high coverage (≥ 6x) and a read length of 75bp, therefore we used DESeq for its broad consensus in the NGS (Next Generation Sequencing) field as a well-grounded tool in this type of DEG pipeline analysis.”

The arguments given by the authors (high coverage, read length) to justify usage DESeq rather than other methods are unsound. High coverage does not eliminate problems caused by lack of biological replicates (i.e. one lacks assessment of variation within class), nor "broad consensus" indicates the suitability of a method in the case of no replication since this has to be explicitly assessed in a methodology comparison work. Anyway, as qPCR-validation is the gold standard to verify differentially expressed genes recovered from any computational algorithm, the authors can just omit this entire statement which seems to be written just to justify DESeq (albeit with the wrong reasons).


2. The authors contradicted themselves with regards to ploidy as an explanation for the increase in histone overexpression. In L.245-247, they wrote that “An increase in the ploidy of MEL-R cell lines might explain the increase in histone gene expression detected by RNA-seq.” However, they later said “These results ruled out the hypothesis that MEL-R tetraploidy was responsible for histone gene over-expression” (L.272 – 273).


3. L.416 – “A genome-wide RNA-seq analysis revealed a general decrease in gene expression in the MEL-R cells compared to the original MEL” – This statement is not correct, because most of the genes (~25000) are not differentially expressed, so how can the authors claim “a general decrease in gene expression”? Rather, I think they mean relatively lower expression values of genes related to actin cytoskeleton pathway. Please rephrase for accuracy.

4. L.29-31 – “These observations revealed that the number of genes expressed in the parental cell line decreased as the cells acquired the resistant phenotype.” And
L.204-205 – “Overall, the total number of genes expressed in MEL decreased as the cells acquired the resistant phenotype.”

These statements are incorrect. The differentially expressed genes are probably getting expressed, even though the FPKM count is 0 (remember, the experiment has no replicates, and low expression counts can easily produce a 0 count by chance). More accurately, it is just that the relative expression of mRNA amount has decreased in the resistant phenotype for some genes (~ 500 out of 25000!)

5. L.426 – 427 - “We speculated that this might be a general mechanism that takes place during cell differentiation.” – I suggest the authors just omit this statement. We only want concrete results in the conclusion section, and speculative material should belong in the Discussion section. Also, “this” is weak and too vague.

6. L.261-264 – “These results implied that the gene expression pattern is heterogeneous during differentiation, suggesting that different genes might be involved in distinct pathways, presumably related to cytoskeleton organization.” – This statement is vague. Of course, it is not surprising that different genes are involved in distinct pathways. Either rephrase for content accuracy or omit.


7. In L.202-204, the authors wrote:

"596 genes with a p-value adjusted less than 0.05 were differentially expressed by more than two-fold between MEL and MEL-R cells, of which 486 genes were up-regulated in MEL cells and 110 were up-regulated in MEL-R cells."

Is this multiple comparison adjustment based on the Benjamini-Hochberg method? The method of p-value adjustment needs to be clearly stated as there are several possibilities (e.g. Benjamini-Hochberg, Benjamini-Yekutieli)


8. L.226 – “a good number of these genes were specific to the hematopoietic lineage ...” – again, the authors were being vague by using “a good number”; how can genes be “specific” to a lineage? The genes are present in all cells, regardless of lineage, the only difference being whether their expression is relatively enhanced or repressed.

Additional comments

There is improvement in the revision, but I am afraid that numerous issues related to English composition as well as lack of accuracy and precision in their statements preclude acceptance of the revision at this point. Also, the abstract on page 4 is different from page 6 of the pdf of the revision. They should be consistent. The abstract may need to be reworded to reflect revisions that arise from this response.

---

## Round 0.3 · Minor Revisions

The manuscript has substantially improved and there are only minor issues left which are detailed in the reviewers' comments.

Reviewer 2 ·

Basic reporting

The reporting of the article has improved. In line 165 of ”tracked changes” manuscript (line 158 in the pdf), replace S3 fig with Fig S3.

Experimental design

No comment.

Validity of the findings

No comment.

·

Basic reporting

In general, the authors have resubmitted an improved revision. A number of minor points remain for their attention.

1. L.28 - 31: “…of which 486 genes were up-regulated in MEL cells and 110 up-regulated in MEL-R cells. These observations revealed that for some genes the relative expression of mRNA amount in the MEL cell line has decreased as the cells acquired the resistant phenotype.”

Since the authors wish to emphasise that the number of genes are down-regulated in MEL-R, they might want to rephrase for more impactful writing:

“…of which 81.5% (486/596) of genes were down-regulated in MEL-R cells. These observations revealed that for many genes the relative expression of mRNA amount in the MEL cell line has decreased as the cells acquired the resistant phenotype.”

2. L.45 - fullstop missing before “These characteristics…”

3. L.53, 54 - “e.g.,” is just “e.g.”

4. L.62 - “developed by Friend and colleagues (Friend et al. 1971)” can be simplified as “developed by Friend et al. (1971)”

5. L.72 “We reported the establishment of an HMBA-resistant cell line (MEL-R) before.” should be “We have previously reported the establishment of an HMBA-resistant cell line (Fernández-Nestosa et al. 2008).” The same reference in L.75 can then be omitted.

6. L.84-94 - The authors still includes their results in the introduction section. They should just focus on introducing the role of the histone proteins in the context of their problem.

7. L.100 - Since the authors know the year, just add this information when the cell lines were obtained from Arthur Skoultchi.

8. L.136 – “2 μg of isolated RNA was” – plural form should be used here

9. L.158 - “S3 Fig.” – should be “Fig. S3”

10. L.193 - add the accession number GSE83567 in the description, for clarity.

11. L.199-200 - “We took a genetic approach to identify potential genes involved in HMBA resistance by using RNA-seq to compare the transcriptomes of MEL and MEL-R cells.”

This statement does not serve any useful purpose in the results section and should be omitted (it is already obvious from the title of the authors' work that they are using RNA-seq which is applied onto the transcriptomes)

12. L.209 -212 “We used DESeq as a complementary differential gene expression analysis to compare the results with those obtained by Cuffdiff (Fig S2). DESeq was employed as an additional DEG calling method since it performs a different expression test (Fishers' exact test) than Cuffdiff (t-test).”

This description should have been moved to RNA isolation and RNA-Seq analysis under
the methods section. Again, I stress that putting methods description in the results section simply distracts from focusing on the results obtained.

13. L.227 – “From the” should be “Among the”

14. L.244 “Those genes, marked with a red asterisk in Table 1 were selected as they showed highest differential expression values, …” should be “Those genes (marked with red asterisk in Table 1), which were selected …”

15. L.260 “, although the difference …” should be “. The difference …”

16. L.267-273 - “We have previously demonstrated that Sfpi1/PU.1 silencing in MEL-R cells is caused by methylation of nearby CpG islands at its promoter (Fernández-Nestosa et al. 2013). Moreover, .... by bisulfite sequencing.”

This paragraph should be moved to the Bisulfite sequencing subsection in the methods section.

17. In the rebuttal letter, the authors said: "The total number of genes is 25791 and they are already described at the GEO database (https://www.ncbi.nlm.nih.gov/geo/query/acc.cgi?acc=GSE83567)"

It is a simple act to insert the number of genes in the text to give readers this immediate information rather than expect them to look up Genbank?

18. L.295-300 “The actin cytoskeleton is composed of an extensive variety of actin regulators and nucleators that interact through a complicated protein network (Moulding et al. 2013; Bezanilla et al. 2015). ..... whether actin was affected by the silencing of genes related to actin polymerization and/or regulation, we evaluated its protein expression by Western blotting”

The authors mixed some methods description in the results section, which I understand is because they approached the topic in spirit of describing the sequence of discoveries. Still, the authors can and should mention the use of Western blotting in the methods section as a means to validate protein expression qualitatively (for whatever result they might observe).

19. L.334 – the gene names are not italicised. Please check again throughout text for any other potential misses.

20. L.407-408 – “MEL-R cells provide a useful model to study the mechanisms that tumor cells use to avoid drug effects and develop resistance.”

This statement adds no useful information and should be omitted for brevity

21. L.409-401”Among the differentially expressed, " - add "genes after "expressed"

22. The sentence o L417 should follow after L.416

23. The authors should acknowledge all the reviewers, and the editor involved in the review of their work, in order to reflect accurately the process of how they arrived at the final version of the manuscript.

24. Caption for Fig 2: 2C, 4C, 8C in subpanel (C) are confounded, so the authors should explain in the caption what 2C, 4C and 8C are. Please note that the x-axis label is missing so the authors should explain what the x-axis is in the caption.

25. Check the caption for Fig4, There is no description for sub panel (A).

26. Fig 5,6 - check that gene names are italicised

27. Does the first statement in L377 follows from the last statement in L376?

28. There paragraphs are still as chunky as ever! Please, have pity on us the readers, and chop them up to ease our reading experience. Examples: start as new paragraph for:

L.382, L.393, L. 351, L.363

The authors should scan through their text again and make appropriate choppings where necessary.

29. The authors should signpost their discussion section using subsections to help readers
grasp important points more easily.

30. Fig S3 - x and y-axis labels (what are the units of measurement?) are missing

31. L.408-409 – “A genome-wide RNA-seq analysis revealed differences in gene expression in the MEL-R cells compared to the original MEL.” should be

“A genome-wide RNA-seq analysis revealed that a subset of genes had significantly lower levels of expression in MEL-R cells compared to MEL cells.”

32. L.214 – “…to MEL-R cells, strengthening RNA-seq data.” – the data is given, one can hardly “strengthen” it. Please rephrase.

Experimental design

No more important issues, except the following one:

1. L.123 - “The quality score (Q) was > 30, performed by the FASTX-Toolkit
(http://hannonlab.cshl.edu/fastx_toolkit/index.html)."

I am not sure why the authors didn’t bother to update this statement. It is grammatically problematic, and furthermore, does not reflect what is observed in the boxplots of quality scores over all sites (which by the way, should be included in the supplemental section)- namely, the median quality score was > 30 across all sites of the reads.

Validity of the findings

Overall I am satisfied with the degree of the reported findings being supported by data, and the previously raised issues have been addressed in the rebuttal letter or corrected by the authors. A few more minor points remain:

1. L.285 - “Quantitative RT-PCR analysis revealed that the level of expression of Dnmt1, the
maintenance methylase enzyme, was higher in MEL-R cells than in …”

Did the authors forget to indicate whether the result was statistically significant?

2. L.369-371 “only 110 from 596 differentially-expressed genes were up-regulated in MEL-R cells. This indicates a tendency towards a general shut-down of gene expression in resistant cells, a situation comparable with what occurs during cell differentiation.”

"General shut-down" does not seem an appropriate description (if something is shutdown, nothing is produced). Please find a more suitable term to describe. Also, the authors should focus on the right context (a glass is half empty vs a glass is half full situation), which seems to be on the 496 differentially genes that are down-regulated in MEL-R, rather than the 110 that are up-regulated, if they wish to emphasize "shut-down". Of course, both are equivalent, but remember they present different reading psychological experience to the reader.

3. In their rebuttal letter, the authors said: "When we said “that the results in both cases follow the same pattern” we referred to the fact that with both methodologies the expression of the majority of the differentially expressed genes is higher in MEL than in MELR cells (96% and 82% in DESeq and Cuffdiff, respectively).Nevertheless, we rephrase this sentence for: …We observed that in both cases the majority of the differentially expressed genes are up-regulated in MEL compare to MEL-R cells… L. 210"

How many differentially expressed genes were found using DESeq, and how many are common with those found using CuffDiff? This is just a simple statistic that the authors can check from their reanalysis, and I am not sure why it cannot be provided in the paper for clarity. They simply need to find out the gene sets detected using DESeq and CuffDiff, and find the intersection set. Surely, this can be done?

---

## Round 0.4 · accepted · Accept

All the previous issues have been properly addressed and the manuscript can now be accepted for publication.